# Partial Lagrangian for Efficient Extension and Reconstruction of Multi-DoF Systems and Efficient Analysis Using Automatic Differentiation

Takashi Kusaka [1,*,†] and Takayuki Tanaka [2,†]

1   Independent Researcher, Sapporo 063-0867, Japan
2   Graduate School of Information Science and Technology, Hokkaido University, Sapporo 060-0814, Japan
*   Correspondence: kusaka@frontier.hokudai.ac.jp
†   These authors contributed equally to this work.

**Abstract:** In the fields of control engineering and robotics, either the Lagrange or Newton–Euler method is generally used to analyze and design systems using equations of motion. Although the Lagrange method can obtain analytical solutions, it is difficult to handle in multi-degree-of-freedom systems because the computational complexity increases explosively as the number of degrees of freedom increases. Conversely, the Newton–Euler method requires less computation even for multi-degree-of-freedom systems, but it cannot obtain an analytical solution. Therefore, we propose a partial Lagrange method that can handle the Lagrange equation efficiently even for multi-degree-of-freedom systems by using a divide-and-conquer approach. The proposed method can easily handle system extensions and system reconstructions, such as changes to intermediate links, for multi-degree-of-freedom serial link manipulators. In addition, the proposed method facilitates the derivation of the equations of motion-by-hand calculations, and when combined with an analysis algorithm using automatic differentiation, it can easily realize motion analysis and control the simulation of multi-degree-of-freedom models. Using multiple pendulums as examples, we confirm the effectiveness of system expansion and system reconstruction with the partial Lagrangians. The derivation of their equations of motion and the results of motion analysis by simulation and motion control experiments are presented. The system extensions and reconstructions proposed herein can be used simultaneously with conventional analytical methods, allowing manual derivations of equations of motion and numerical computer simulations to be performed more efficiently.

**Keywords:** Lagrange equation; automatic differentiation; divide-and-conquer approach





## 1. Introduction

In dynamics, the Newton–Euler and Lagrange methods are well known for formulating and calculating the equations of motion and used for different purposes [1–4]. The Newton–Euler method is a computational procedure that can be used in multi-degree-of-freedom (DoF) models with the systematic algorithm for one DoF at a time [5–7]. Therefore, it is a suitable method for obtaining numerical solutions, but not for analytical solutions. Conversely, the Lagrange method is a formulation procedure that yields analytical solutions that can be understood as physically meaningful terms. However, the amount of computation is enormous when analyzing multi-DoF systems. Therefore, it is generally used for the analysis of systems with fewer DoF.

Numerical solutions that can be computed with the Newton–Euler method are useful for implementing equipment that uses equations of motion, such as real-time feedback control of power assistance in human–machine coordination systems in the field of robotics [8,9]. However, when designing algorithms or conducting theoretical analysis of systems, the meaning of each physical term in the analytical solution obtained by the Lagrange method is extremely important. For example, it is used in the behavior analysis

and control of walking robots and drones [10–13]. We use these methods for different applications. In our past research, we used the Lagrange method for the analysis of human body motion modeled in four DoF and the design of control systems [14–16]. We also used the Newton–Euler method to analyze the dynamics of a 7-link (21-DoF) spinal column model for a wearable system [17,18].

A major problem with the Lagrange method is that the amount of computation increases explosively when the number of DoF increases, and the results become complicated. While computer algebra systems can be used to perform large-scale system analysis, manual calculations are extremely difficult to handle even with four DoF, and calculation errors are likely to occur. Because the equations of motion with multi-DoF are extremely complex, many techniques to generate them automatically by computer have been studied. In fact, the equations of motion for the robot arm can be obtained with mathematical processing software, such as Mathematica's Robotica [19] or MatLab's TMTDyn [20], and calculations such as the TMT method are used to optimize the computer calculations [21]. However, the results of large systems collectively are very complex and not reusable.

Therefore, we propose to solve this problem by introducing a partial Lagrangian and postural operator. The partial Lagrangian uses the divide-and-conquer approach [22,23] to divide the equations of motion, which become complex when the number of DoF increases, into the smallest units. The use of the partial Lagrangian reduces the computational load because the terms in the equations of motion for increasing DoF can be treated independently. In other words, a similar process of division can be performed for each DoF, and the results integrated to obtain an exact analytical solution. This has two advantages. First, when the system is extended or reconstructed, only the affected part of the system needs to be calculated, making this analysis method flexible in terms of system configuration. The second advantage is that the modularization of the calculation unit minimizes the burden of obtaining the equations of motion by manual calculation. The modularized calculation is also compatible with computer processing because it involves iterative calculations and the final integration of similar processes.

Studies analyzing multibody dynamics using the divide-and-conquer algorithm (DCA) include the generalized DCA [24], DCAe [25], and a study of sensitivity analysis using DCA [26]. The generalized DCA is an extension method of the DCA for modeling constrained multibody systems, the DCAe is a reconstruction of DCA for efficient handling of multibody dynamics, and the sensitivity analysis using DCA treats the DCA as a critical tool for efficient analysis of multibody dynamics. All of these are based on dividing the generalized force in a binary tree and applying the divide-and-conquer method. Therefore, mechanical constraints are important in all of these studies. In other past studies, such as those mentioned above, no method has focused on Lagrangian dividing. Since the proposed method applies to the Lagrangian (energy), which is abstract, it can be applied to any system that can be described by a Lagrangian that is linearly independent.

Recursive algorithms, such as an algorithm applying the Gibbs–Appel equation [27] and the harmony search algorithm [28], are also known to be very effective for complex systems, such as parallel robots. Our approach differs from recursive algorithms in that it exploits the linear independence of the Lagrangian for partitioning and reuse of computed results. As a benefit of partitioning, we can derive analytical solutions and gain computational efficiencies in computerized numerical solutions. Therefore, the final result of the proposed method is completely equal to the usual Lagrangian method and can be used without the need for a recursive algorithm.

We propose a numerical analysis method based on automatic differentiation as the optimal analysis method for the partial Lagrangian. Automatic differentiation is a method used for training neural networks [29–34] We also confirm that the proposed partial Lagrangian and automatic differentiation can be used to simulate multi-linked manipulators easily and that the system can be easily extended and reconstructed.

## 2. Methods

### 2.1. Partial Lagrangian

This section provides an overview of the partial Lagrangian. Specific examples are given in the next section. The usual Lagrangian requires the total kinetic and potential energy of the system, so the term explodes with increasing DoF. There is a known efficient method of recursive computation using the linear independence of the Lagrangian [35,36]. We extend this idea and consider how to design a more efficient system using the divide-and-conquer approach by organizing it in units of the partial Lagrangian.

First, the Lagrangian $\mathcal{L}$ of the $n$-DoF system is the difference between the total kinetic energy $K$ and the total potential energy $P$, as follows

$$\mathcal{L} = K - P \tag{1}$$

Here, if we decompose each type of energy into DoF using the distributive property, we can transform it as follows.

$$\mathcal{L} = \sum_{i=1}^{n} K_i - \sum_{i=1}^{n} P_i = \sum_{i=1}^{n}(K_i - P_i) = \sum_{i=1}^{n} \mathcal{L}_i \tag{2}$$

This $\mathcal{L}_i$ is defined as a partial Lagrangian. The subscript $i$ denotes the division into partial Lagrangians. An $n$-DOF system will have n partial Lagrangians, corresponding to each DoF as $i = 1, 2, \cdots, n$. Since the Lagrangian $\mathcal{L}$ is linearly independent, the partial Lagrangian $\mathcal{L}_i$ for the $i$-th DoF consists of the partial kinetic energy $K_i$ and the partial potential energy $P_i$.

Considering the generalized coordinate $q_k$, the equation of motion with the Lagrangian is as follows:

$$\frac{d}{dt}\left(\frac{\partial}{\partial \dot{q}_k}\mathcal{L}\right) - \frac{\partial}{\partial q_k}\mathcal{L} = \tau_k \tag{3}$$

where the subscript $k$ represents the $k$-th equation of motion and $k \leq n$ for the $n$-DoF system. $\tau_k$ represents the generalized force of $k$-th DoF since the above equation is the usual Lagrangian equation of motion. For simplicity, let the differential operator on the left-hand side be defined formally as $D_k = \frac{d}{dt}\frac{\partial}{\partial \dot{q}_k} - \frac{\partial}{\partial q_k}$. In other words, the equation of motion for the $k$-th DoF is $D_k\mathcal{L} = \tau_k$.

Here, considering the partial Lagrangian $\mathcal{L}_i$,

$$D_k\mathcal{L} = D_k \sum_{i=1}^{n} \mathcal{L}_i = \sum_{i=1}^{n}(D_k\mathcal{L}_i) = \sum_{i=1}^{n} \tau_{ki} \tag{4}$$

because the order of the sum and derivative can be exchanged using term-wise differentiation. This $\tau_{ki}$ is defined as the partial generalized force.

Now consider the components of $\mathcal{L}$. If $i \geq k$, $\mathcal{L}_k$ contains $q_i$, but if $i < k$, $\mathcal{L}_k$ does not contain $q_i$. Therefore, the partial generalized force $D_k\mathcal{L}_i$ is as follows.

$$D_k\mathcal{L}_i = \begin{cases} \tau_{ki} & (i \geq k) \\ 0 & (i < k) \end{cases} \tag{5}$$

This can be summarized as shown in Table 1. Looking at this table, the equation of motion $D_k\mathcal{L} = \tau_k$ using the Lagrangian method corresponds to calculating all of the entries in row k simultaneously. Therefore, the calculation explodes as the number of DoF increases. However, the partial Lagrangian $D_k\mathcal{L}_i = \tau_{ki}$ is equivalent to splitting this calculation and performing the calculation with respect to $\mathcal{L}_i$. In other words, the analysis is column-wise. Therefore, the final equivalent analytical solution is obtained by summing, but it can be computed by dividing the solution in order, starting from $i = 1$.

In addition, the calculation results of $\mathcal{L}_{(i-1)}$ can be diverted for the calculation of $\mathcal{L}_i$, thus reducing the amount of the calculation. Furthermore, in the case of the Lagrangian equation $D_k\mathcal{L} = \tau_k$, if the robot's link is extended or the number of DoF is changed after the analytical solution is obtained, all calculations must be redone. Conversely, when the partial Lagrangian is used, the calculation is independent for each DoF, so the results of the root side calculation can be reused. Therefore, by managing the system in units of partial Lagrangian modules, system extensions, and changes can be handled in a prepared manner.

For example, to extend the system, simply add a new term regarding $\mathcal{L}_{(n+1)}$ as follows, and the summing part can be reused.

$$\text{Current system: } \tau_k = \sum_{i=1}^{n} D_k\mathcal{L}_i \tag{6}$$

$$\downarrow$$

$$\text{Extended system: } \tau_{k,\,\text{new}} = \underbrace{\sum_{i=1}^{n} D_k\mathcal{L}_i}_{\text{Reusable term}} + \underbrace{D_k\mathcal{L}_{(n+1)}}_{\text{New term by the extension}} \tag{7}$$

In addition, system changes can be realized in the same way.

$$\text{Current system: } \tau_k = \sum_{i=1}^{n} D_k\mathcal{L}_{i,\,\text{old}} = \sum_{i=1}^{n-1} D_k\mathcal{L}_i + D_k\mathcal{L}_{n,\,\text{old}} \tag{8}$$

$$\downarrow$$

$$\text{Changed system: } \tau_{k,\,\text{new}} = \sum_{i=1}^{n} D_k\mathcal{L}_{i,\,\text{new}} = \underbrace{\sum_{i=1}^{n-1} D_k\mathcal{L}_i}_{\text{Reusable term}} + \underbrace{D_k\mathcal{L}_{n,\,\text{new}}}_{\text{Replaced term}} \tag{9}$$

**Table 1.** Partial Lagrangian vs. Lagrange method.

| | | | | **Partial Lagrangian** | | | | | **Lagrange Method** |
|---|---|---|---|---|---|---|---|---|---|
| | $\mathcal{L}_1$ | $\mathcal{L}_2$ | $\mathcal{L}_3$ | $\cdots$ | $\mathcal{L}_i$ | $\cdots$ | $\mathcal{L}_n$ | $\xrightarrow{\Sigma}$ | $\mathcal{L}$ |
| $D_1$ | $\tau_{11}$ | $\tau_{12}$ | $\tau_{13}$ | $\cdots$ | $\tau_{1i}$ | $\cdots$ | $\tau_{1n}$ | $\rightarrow$ | $\tau_1$ |
| $D_2$ | $0$ | $\tau_{22}$ | $\tau_{23}$ | $\cdots$ | $\tau_{2i}$ | $\cdots$ | $\tau_{2n}$ | $\rightarrow$ | $\tau_2$ |
| $D_3$ | $0$ | $0$ | $\tau_{33}$ | $\cdots$ | $\tau_{3i}$ | $\cdots$ | $\tau_{3n}$ | $\rightarrow$ | $\tau_3$ |
| $\vdots$ | $\vdots$ | $\vdots$ | $\vdots$ | $\ddots$ | $\vdots$ | | $\vdots$ | $\vdots$ | $\vdots$ |
| $D_k$ | $0$ | $0$ | $0$ | $\cdots$ | $\tau_{ki}$ | $\cdots$ | $\tau_{kn}$ | $\rightarrow$ | $\tau_k$ |
| $\vdots$ | $\vdots$ | $\vdots$ | $\vdots$ | | $\vdots$ | $\ddots$ | $\vdots$ | $\vdots$ | $\vdots$ |
| $D_n$ | $0$ | $0$ | $0$ | $\cdots$ | $0$ | $\cdots$ | $\tau_{nn}$ | $\rightarrow$ | $\tau_n$ |

In summary, when the partial Lagrangian is generalized to $n$ DoF, the following procedure can be used to calculate the partition.

1.  Dynamics: Differentiation of state variables as physical constraints.
2.  Kinematics: Position $p_i$ and velocity $\dot{p}_i$ as geometric constraints.
3.  Partial Lagrangian.

    (a)  Quadratic form: calculate $p_i^T p_i$ and $\dot{p}_i^T \dot{p}_i$ to find the energies. Calculate the partial energies: $K_i$ and $P_i$.

    (b)  Compute the partial Lagrangian: $\mathcal{L}_i$.

4.  Find the partial generalized force: $D_k\mathcal{L}_i = \tau_{ki}$. If it is a multi-degree-of-freedom system, find the sum $\tau_k = \sum_{i=1}^{n} \tau_{ki}$.

This calculation procedure is performed in order from the smallest to the largest value of $i$, and finally, the equation of motion can be derived by computing the sum. Therefore, the complex multi-DoF Lagrangian equations can be obtained with relative ease using a divide-and-conquer approach. Censoring in the middle is equivalent to the equation of motion for a short robot arm, and it is also easy to add $i = n + 1$ later.

### 2.2. Postural Operator for Hand Calculations

By using the partial Lagrangian, the Lagrangian is divided and treated independently with respect to each DoF. In this subsection, we define a posture operator for more efficient computations when dealing with partial Lagrangians. Let the attitude operator in the planar coordinates be a vector of length 1, such that

$$e_\theta = \begin{bmatrix} \cos \theta \\ \sin \theta \end{bmatrix} \tag{10}$$

This has the following trivial properties:

- Cancellation: $e_\theta^T e_\theta = 1$
- Interference: $e_{\theta_{12}}^T e_{\theta_1} = e_{12}^T e_1 = \cos \theta_2$
- Derivative: $\frac{d}{dt} e_\theta = \begin{cases} \dot{\theta} e_{\theta + \frac{\pi}{2}} = \dot{\theta} e'_\theta & \text{if } \theta = \theta(t) \\ 0 & \text{if } \theta = \text{const.} \end{cases}$

where $\theta_{12} = \theta_1 + \theta_2$ and $e_{\theta_i} = e_i$ for notational simplicity. The phase difference of $+\frac{\pi}{2}$ is defined as $e_{\theta + \frac{\pi}{2}} = e'_\theta$.

This is similar to the stationary phasor [37–39] in electrical engineering, which uses a polar form to improve the perspective of the formula expansion. Here, we treat it as a vector rather than a complex form. This also makes it intuitively consistent with the orthogonal form expression expansion. The description of motion in a two-dimensional plane is simplified by using the above properties. In the three-dimensional case, a similar argument can be made using a versor [40–43] with a magnitude of 1 as a rotation by quaternions or using a rotation matrix [44]. We deal with the two-dimensional case because time is used as the third axis for the visualization of the simulation results.

### 2.3. System Analysis Using Automatic Differentiation for Numerical Calculation

Next, we consider the numerical analysis using the partial Lagrangian. The advantage of the partial Lagrangian is that extensions and reconstructions of the system can easily be realized by splitting the Lagrangian. If the equations are transformed to the form of linear differential equations for each DoF for dynamics analysis, this is equivalent to the usual Lagrangian and loses the advantage of the split calculation.

An analytical method that can directly handle $D_k \mathcal{L}_i = \tau_i k$, which is a formal description using the partial Lagrangian, is desirable. Therefore, we use an analysis method that applies automatic differentiation. The solution of the equation of motion by the partial Lagrangian is obtained by generating a computational graph by automatic differentiation and performing a gradient calculation using the balance of the partial differential equation as a constraint. Applications to the analysis of partial differential equations other than neural networks are used in the fields of rigorous simulations, such as the finite element method analysis [45,46], dynamics calculations [47,48], and electronic circuit analysis [49–51].

An example of dynamics computed by automatic differentiation is shown in Appendix A. As indicated in the function in the Appendix, the program for generating a graph of differential equation calculations for automatic differentiation is written in the following flow.

1. Dynamics: The time evolution of the state variable is registered in the calculation graph as a constraint.
2. Kinematics: Register a geometric constraint on a calculation graph.
3. Register a partial Lagrangian in the computed graph
4. Register the partial generalized force, and if it is a multi-DoF, find the sum for each DoF.

5.　　Backpropagate and find the coefficients as the gradient of the state variable.

This can be described in exactly the same flow as the calculation algorithm for the partial Lagrangian method presented in Section 2.1. Therefore, by using automatic differentiation and calculating dynamics by the partial Lagrangian method, the calculation of the equations of motion can be realized using a divide-and-conquer approach. In addition, each calculation is divided and modularized, allowing the system to be reconstructed instantly.

## 3. Results

We present some examples and simulations to confirm the effect of the partial Lagrangian. For simplicity, we assume an n-link manipulator moving on a plane and describe it using the postural operator. Although the pendulum model is assumed for the simplicity of explanation, rigid body links can be treated in the same way as long as the Lagrangian is a linear sum. The link parameter for link i has a link length of $l_i$, mass $m_i$, and stiffness $k_i = 0$. The link length is read as the total length when dealing with connections between links or the length to the center of gravity when dealing with the center of gravity, as appropriate.

### 3.1. Example of 1-DoF: Effect of Postural Operator

In the case of one DoF, $\tau_{11} = D_1\mathcal{L}_1 = D_1\mathcal{L} = \tau_1$; thus, the partial Lagrangian result is exactly equal to the usual Lagrangian method. Here, we use the single pendulum [52] as an example of a rotational joint. This calculation is very simple, but to confirm the effect of the postural operator and the flow of the partial Lagrangian processing, we show it in detail as an example.

First, as a geometric constraint, the link end position and its velocity are as follows from the kinematics.

$$\boldsymbol{p}_1 = l_1\boldsymbol{e}_1 \tag{11}$$

$$\dot{\boldsymbol{p}}_1 = l_1\frac{d}{dt}\boldsymbol{e}_1 = l_1\dot{\theta}_1\boldsymbol{e}_1' \tag{12}$$

These quadratic forms are then obtained as inner products.

$$\boldsymbol{p}_1^T\boldsymbol{p}_1 = l_1^2 \underbrace{\boldsymbol{e}_1^T\boldsymbol{e}_1}_{\text{cancellation}} = l_1^2 \tag{13}$$

$$\dot{\boldsymbol{p}}_1^T\dot{\boldsymbol{p}}_1 = l_1^2\dot{\theta}_1^2 \underbrace{\boldsymbol{e}_1'^T\boldsymbol{e}_1'}_{\text{cancellation}} = l_1^2\dot{\theta}_1^2 \tag{14}$$

From these, the kinetic energy $K$ and potential energy $P$ can be calculated as follows.

$$K_1 = \frac{1}{2}m_1\dot{\boldsymbol{p}}_1^T\dot{\boldsymbol{p}}_1 = \frac{1}{2}m_1l_1^2\dot{\theta}_1^2 \tag{15}$$

$$P_1 = \frac{1}{2}k_1\boldsymbol{p}_1^T\boldsymbol{p}_1 + m_1gy_1 = m_1g\sin\theta_1. \tag{16}$$

The partial Lagrangian calculated using these is $\mathcal{L}_1 = K_1 - P_1 = \mathcal{L}$. Therefore, its equation of motion can be obtained as $\tau_{11} = D_1\mathcal{L}_1 = D1\mathcal{L} = \tau_1$. This should be equivalent to the equation of motion for an inverted pendulum in a general textbook.

To confirm this, we analyze the behavior of the equations of motion using automatic differentiation. For one link, the analysis can be performed using the sample of automatic differentiation shown in Appendix A. The link parameters are $m_1 = 3$, $k_1 = 0$, $l_1 = 2$, with an appropriate damping term $d_1 = 1$ to make the behavior easier to understand. The results are shown in Figure 1a. Posture control can also be simulated simply by writing a control input to the generalized forces of the equations of motion in the calculation graph. The results of the proportional–derivative (PD) control with a target angle of 110° are shown in

Figure 1b. The PD gains were set to low values ($K_{p1} = 200$ and $K_{d1} = 50$, respectively) to make the behavior easy to understand.

The results show how to calculate the partial Lagrangian by hand for the simplest case and analyze it by automatic differentiation. This is the minimum unit of the divide-and-conquer approach for the Lagrangian, and from the next section, we will confirm that this procedure can be repeated according to Table 1 to obtain the desired equations of motion.

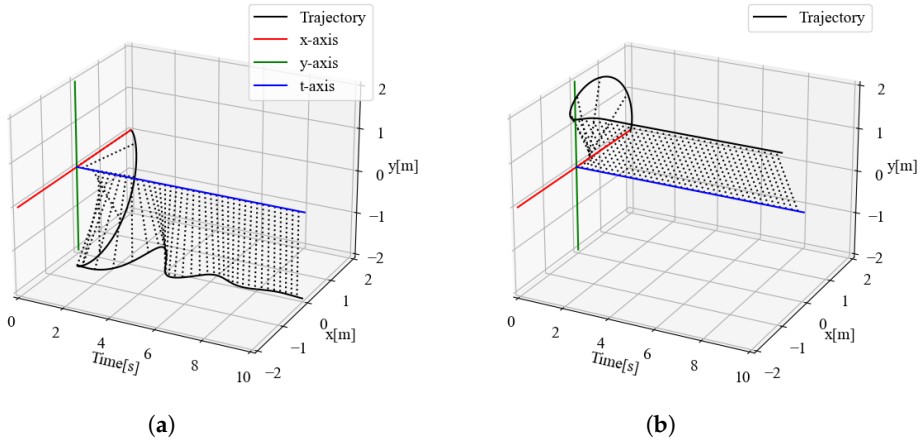

(**a**)                    (**b**)

**Figure 1.** Motion analysis of the 1-link system using automatic differentiation. (**a**) Damping oscillation; (**b**) PD control.

### 3.2. Example of 2-DoF: Effect of Divide-and-Conquer by Partial Lagrangian

In the previous chapter, we confirmed the reduction in computational complexity due to the postural operator $e_\theta$ on a 1-DoF example. Next, we will use a 2-DoF example to confirm this effect. According to Table 1, only $\tau_{12}$ and $\tau_{22}$ need to be added. As a simple example, we consider a double pendulum [53,54], which extends the 1-DoF example.

The same procedure as for the 1-DoF is used to obtain the partial Lagrangian $\mathcal{L}_2$.

First, the geometric constraints are as follows.

$$p_2 = p_1 + l_2 e_{12} = l_1 e_1 + \underbrace{l_2 e_{12}}_{\text{new information}} \tag{17}$$

$$\dot{p}_2 = \dot{p}_1 + l_2 \frac{d}{dt} e_{12} = l_1 \dot{\theta}_1 e_1' + \underbrace{l_2 \dot{\theta}_{12} e_{12}'}_{\text{new information}} \tag{18}$$

Then, their quadratic forms are

$$p_2^T p_2 = l_1^2 \underbrace{e_1^T e_1}_{\text{cancellation}} + l_2^2 \underbrace{e_{12}^T e_{12}}_{\text{cancellation}} + l_1 l_2 \underbrace{e_1^T e_{12}}_{\text{interference}}$$
$$= l_1^2 + l_2^2 + l_1 l_2 \cos \theta_2 \tag{19}$$

$$\dot{p}_2^T \dot{p}_2 = l_1^2 \dot{\theta}_1^2 \underbrace{e_1'^T e_1'}_{\text{cancellation}} + l_2^2 \dot{\theta}_{12}^2 \underbrace{e_{12}'^T e_{12}'}_{\text{cancellation}} + l_1 l_2 \dot{\theta}_1 \dot{\theta}_{12} \underbrace{e_1'^T e_{12}'}_{\text{interference}}$$
$$= l_1^2 \dot{\theta}_1^2 + l_2^2 \dot{\theta}_{12}^2 + l_1 l_2 \dot{\theta}_1 \dot{\theta}_{12} \cos \theta_2. \tag{20}$$

The kinetic and potential energies consisting of the link parameters for $i = 2$ are

$$K_2 = \frac{1}{2} m_2 \dot{p}_2^T \dot{p}_2 \tag{21}$$

$$P_2 = \frac{1}{2} k_2 p_2^T p_2 + m_2 g y_2 \tag{22}$$

Since this partial Lagrangian is $\mathcal{L}_2 = K_2 - P_2$, the equations of motion are

$$\tau_1 = \tau_{11} + \tau_{12} = \tau_{11} + D_1\mathcal{L}_2 \tag{23}$$

$$\tau_2 = \qquad \tau_{22} = \qquad D_2\mathcal{L}_2, \tag{24}$$

also using the result of $\tau_{11}$ with one DoF. In other words, the calculation was reduced by one previous result, $\tau_{11}$, in finding the equations of motion for two DoF. To obtain the equation of motion for $n$ DoF, only the $n$ columns of Table 1 must be calculated, and the calculations from columns 1 to $n-1$ are unnecessary because they can be reused. References [53,54] show that the equation of motion of a double pendulum is complicated to be solved during the process if it is obtained by the usual Lagrangian method; however, it can be described in a simple manner by the partial Lagrangian method. This simplifies the calculation even for an increased DoF and prevents calculation mistakes in manual calculations.

We confirm that this result is correct by simulation with automatic differentiation. The simulation can take full advantage of the divide-and-conquer effect of the partial Lagrangian. In the example program shown in Appendix A (Listing A1), to obtain $\mathcal{L}_2$, only two lines of the geometric constraint need to be rewritten according to the system. Therefore, the idea of division of the process in the partial Lagrangian corresponds to the program, and the same calculation results obtained using the complicated equations of motion can also be obtained as an iteration of this module.

The results of the 2-DoF example are shown in Figure 2a as damped oscillations and that of adding the PD control to the joints are shown in Figure 2b. The link parameters are $m_1 = 3$, $k_1 = 0$, $l_1 = 2$, $d_1 = 1$ for the first joint and $m_2 = 1$, $k_2 = 0$, $l_2 = 1$, $d_2 = 1$ for the second. Target values of $\theta_{1ref} = 80°$ and $\theta_{2ref} = 30°$ were used for PD control, and its gains are $K_{p1} = 100$, $K_{p2} = 50$, $K_{d1} = 50$, and $K_{d2} = 20$. It can be shown that the desired behavior can be analyzed even in the case of multi-DoF.

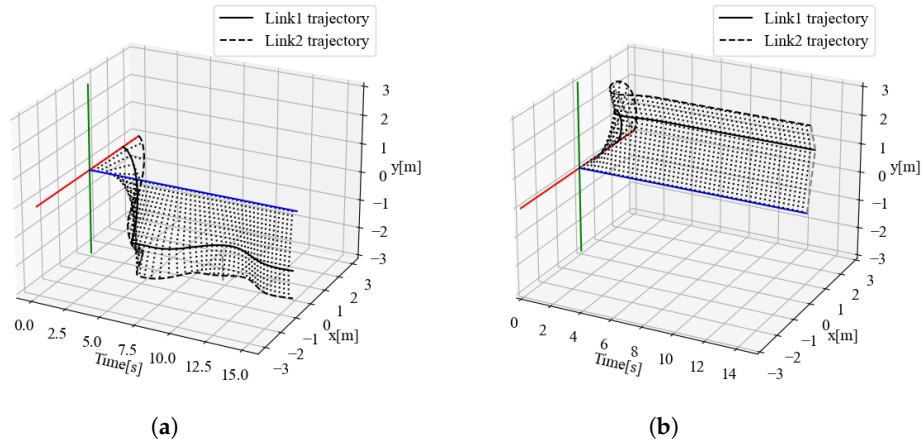

    (a)                                     (b)

**Figure 2.** Motion analysis of a two-link system using automatic differentiation. (**a**) Damping oscillation; (**b**) PD control.

### 3.3. Example of 3-DoF and Changing System Construction

First, we show an example of a triple pendulum [55,56] that extends the 2-DOF example. Then, we look at the system reconstruction with partial Lagrangian when the second link is changed from a rotational joint to a linear motion joint. The calculation of $\mathcal{L}_3$ is an iteration of Equations (8)–(17) and is abbreviated here. The required equations of motion are as follows.

$$\tau_1 = \tau_{11} + \tau_{12} + \tau_{13} = \tau_{11} + \tau_{12} + D_1\mathcal{L}_3 \tag{25}$$

$$\tau_2 = \qquad \tau_{22} + \tau_{23} = \qquad \tau_{22} + D_2\mathcal{L}_3 \tag{26}$$

$$\tau_3 = \qquad \tau_{33} = \qquad D_3\mathcal{L}_3 \tag{27}$$

Since the only new term that needs to be calculated is the term related to $\mathcal{L}_3$, we can extend the link easily even by hand calculation, by using the postural operator as in the case up to the 2-DoF example. References [55,56] show that the Lagrangian becomes very complex with three DoF; thus, this approach has the advantage of performing the partitioning calculation with the partial Lagrangian.

This behavior is confirmed by simulation with automatic differentiation. The results of the damped oscillation are shown in Figure 3a, and the results with the PD control are shown in Figure 3b. The target angles for PD control are $\theta_{1ref} = 125°$, $\theta_{2ref} = -30°$, and $\theta_{3ref} = -30°$, and its gains are $K_{p1} = 1000$, $K_{p2} = 1000$, $K_{p3} = 500$, $K_{d1} = 200$, $K_{d2} = 200$, and $K_{d3} = 100$. As with the 2-DoF case, this simulation can be handled simply by modifying two lines of the geometric constraint for $\mathcal{L}_3$; thus, the implementation cost is very small.

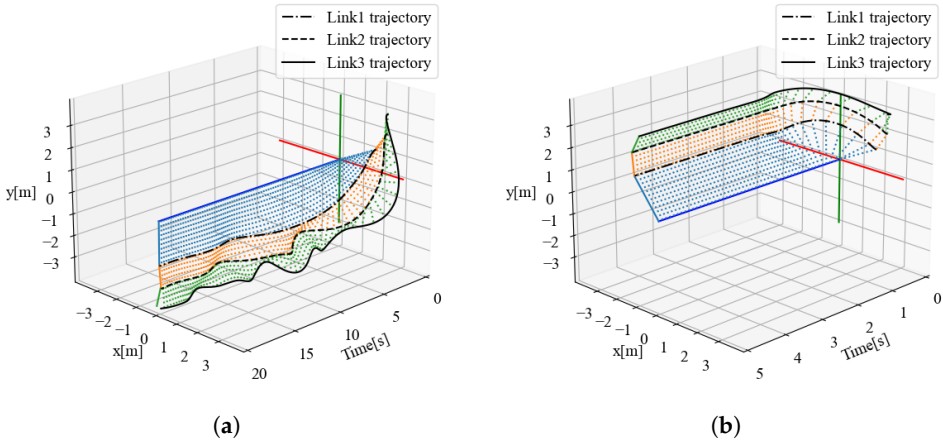

**Figure 3.** Motion analysis of the 3-link system using automatic differentiation. (**a**) Damping oscillation; (**b**) PD control.

Next, as an example of system reconstruction, a simulation of a system in which the second link is changed to a linear joint is shown in Figure 4a,b. The second link is controlled to shorten its length by the PID control, and the other joints are the damped vibrations. The initial length of the second link is $l_2 = 1$ and its target length is $l_{2ref} = 0.1$. The results show that the desired behavior can be achieved even when the intermediate link is changed as an example of the system reconstruction. As mentioned above, in the automatic differentiation program, the simulation can be realized merely by rewriting the time-varying parameters of the geometric constraint.

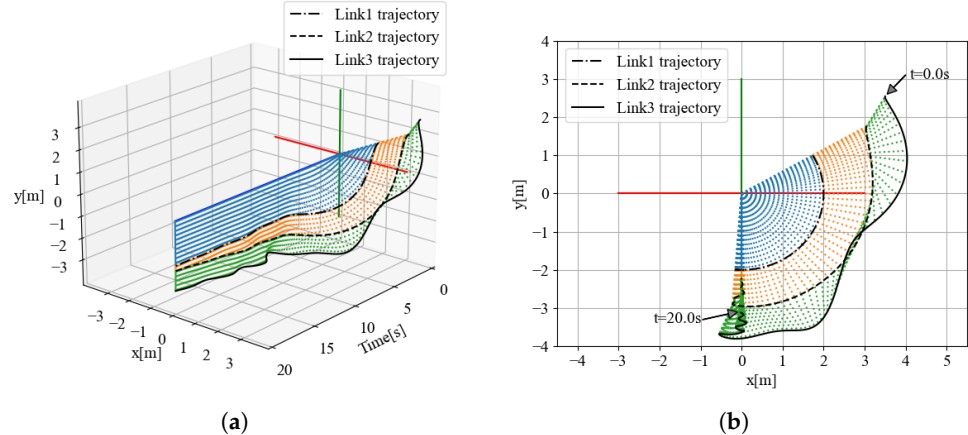

**Figure 4.** System reconstruction example (link 2 is changed from rotational joint to linear slider). (**a**) Side view; (**b**) Front view.

## 4. Discussion

In this section, we discuss the advantages of the proposed partial Lagrangian. As mentioned above, the purpose of the usual Lagrangian is to derive an analytical solution; thus, its application to multiple degrees of freedom is difficult. However, by dividing it into partial Lagrangians, it can be handled relatively easily even with multiple degrees of freedom, and can be used to compute numerical solutions. In this section, we discuss computational complexity and scalability by considering extensions up to 10 links, as shown in Figure 5.

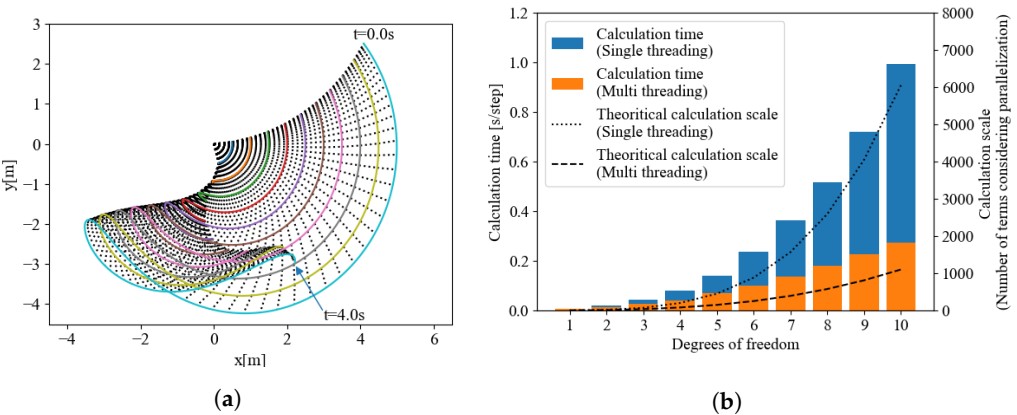

|        (a)        |        (b)        |

**Figure 5.** Experimental results of a large-scale model using the partial Lagrangian . (**a**) Damping oscillation by the 10-DoF pendulum; (**b**) computational cost of automatic differentiation (all simulations were performed by Intel Core i5-10400 CPU @2.90 GHz).

### 4.1. Computational Advantages of Partial Lagrangian

The usual Lagrangian is generally difficult to use with the multi-DoF system because the number of terms explodes with increasing degrees of freedom. The partial Lagrangian uses the divide-and-conquer approach to divide the computation of the Lagrange equation into its smallest module, $D_k \mathcal{L}_i = \tau_{ik}$. This makes it relatively easy to obtain an exact analytical solution eventually by repeating the same simple procedure, even if the number of DoF is increased.

We consider the advantages of the partial Lagrangian from two perspectives: the derivation of the equations of motion by hand calculations and computer simulations.

First, for manual calculations, the introduction of the postural operator together with the partial Lagrangian simplifies the calculation with respect to rotational joints and significantly reduces the number of calculations compared to those that would otherwise be required. In addition, divide-and-conquer by the partial Lagrangian prevents calculation errors due to manual calculations because the number of calculations is small for each module.

Second, for computer simulations, the gradient calculation of the calculation graph by automatic differentiation was used as an efficient method to process the formal description of the divide-and-conquer approach by the partial Lagrangian, $D_k \mathcal{L}_i = \tau_{ik}$. Automatic differentiation is a different method of analysis from mathematical and numerical differentiation. Automatic differentiation can solve the inefficiency of mathematical differentiation and the accuracy problem of numerical differentiation. It is compatible with partial Lagrangians because it can describe partial differential equations directly and perform the calculation of the balance. The partial Lagrangian allows for efficient simulation by iterating the modularized process through automatic differentiation and integrating it at the necessary stages.

As an example of this, a simulation in which the model is extended to 10 links is shown in Figure 5a. Here, we consider Figure 5b as an efficiency improvement by dividing it into partial Lagrangians. By dividing, computers can compute independent components

using multi-threading. The order of computational amount of $n$-DoF without dividing can be estimated as the number of terms as follows. The $\mathcal{O}$ means Landau symbol.

$$\underbrace{\frac{1}{2}(n^2 + n)}_{\text{Number of all } \tau_{ki}} \times \underbrace{(n^2 + 2n)}_{\text{Number of terms in } \tau_{ki}} = \mathcal{O}(n^4) \tag{28}$$

The fact that the computational cost is $\mathcal{O}(n^4)$ in the analysis of general dynamics is stated in the reference [26] and supports this result.

On the contrary, using partial Lagrangian, the generation of the computational graph and the gradient calculation (i.e., torque calculation) with multi-threading is shown below.

$$\underbrace{n}_{\substack{\text{Max number of dividing} \\ \text{into partial Lagrangian}}} \times \underbrace{(n^2 + 2n)}_{\text{Number of terms in } \tau_{ki}} = \mathcal{O}(n^3) \tag{29}$$

As shown above, the division into partial Lagrangians allows the linearly independent parts to be computed in parallel, lowering the order of computation from $\mathcal{O}(n^4)$ to $\mathcal{O}(n^3)$. These computational amounts are consistent with the results of the actual simulation shown in Figure 5b.

### 4.2. Application of Partial Lagrangian as an Extension or Restructuring of the System

Another advantage of the partial Lagrangian is that it can handle system extensions and reconstructions in units of divided modules. This is due to the linear independence of each column of Table 1. The pendulum examples shown in Section 3 show the flow of the system extension from one to three DoF. The system extended to 10-DoF to evaluate the computational amount in the previous section is an extension of the 3DoF model with additional link parameters, all computed with the same partial Lagrangian module.

With the partial Lagrangian, the calculations required for energy and partial derivatives are about half of those of the usual Lagrangian. This makes it easier to handle the multi-DoF system than the usual Lagrangian, but the difficulty increases with the DoF number. Moreover, if the middle link is changed, it is necessary to go back that far in Table 1. However, in the case of simulation with automatic differentiation, energy calculation and partial differentiation are performed automatically, so system reconstruction, such as the intermediate link changes, can be performed merely by rewriting the geometric constraint part. Therefore, similar to the conclusion of the computational advantage, the divide-and-conquer approach using the partial Lagrangian and module-by-module processing using an automatic differentiation algorithm allows for the easy dynamics analysis of complicated systems and their configuration changes with only a formal description of the partial Lagrangian.

## 5. Conclusions

We proposed the partial Lagrangian to handle the Lagrange equation efficiently using a divide-and-conquer approach. The partial Lagrangian makes it possible to handle extensions and reconstructions of the system easily to obtain analytical solutions, which was not previously possible. In addition, the introduction of the postural operator together with the partial Lagrangian facilitates the derivation of the equations of motion by hand calculation. The division of computations by the partial Lagrangian and reduction of computational complexity by the postural operator reduce the computational cost of manual calculations of energy and partial derivatives, even with multi-DoF.

Furthermore, we proposed a numerical method of computing the partial Lagrangian using automatic differentiation for simulation. Since automatic differentiation calculates the energy and partial derivatives, it is compatible with the partial Lagrangian formal description, and the system designer needs only to describe the geometric constraints to analyze the motion using the partial Lagrangian's divide-and-conquer approach.

We confirmed that the division into partial Lagrangians allows us to take advantage of multi-threading, reducing the computational complexity that normally requires $\mathcal{O}(n^4)$ to $\mathcal{O}(n^3)$. Even when extended to the multi-DoF system, the numerical solution can be obtained efficiently by repeatedly calling the partial Lagrangian module. As an example of a multi-DoF, we confirmed its effectiveness with a 10-DoF pendulum model.

By replacing or combining the proposed method with conventional systems, the system extensions and simulations can be realized more efficiently than previously possible in the field of design and control using analytical solutions of the conventional equations of motion. However, because the partial Lagrangian uses the linear independence of each DoF to split its computation, the split may not work well if its assumptions change. Several recent studies have shown that recursive algorithmic solutions are also effective in parallel-linked systems [27,28]. Therefore, our future work will include applications to systems other than the serial link manipulators treated in this study, such as closed-link systems and parallel-link systems.

**Author Contributions:** Conceptualization, T.K.; investigation, T.K.; methodology, T.K.; supervision, T.T.; validation, T.K. and T.T.; writing—original draft, T.K.; writing—review and editing, T.T. All authors have read and agreed to the published version of the manuscript.

**Funding:** This research received no external funding

**Institutional Review Board Statement:** Not applicable.

**Informed Consent Statement:** Not applicable.

**Data Availability Statement:** Not applicable.

**Conflicts of Interest:** The authors declare no conflict of interest.

## Appendix A. A Sample Analysis of Partial Lagrangian Using Automatic Differentiation

A sample analysis of an edited differential equation with automatic differentiation is shown below. We begin with a sample program for the case of one degree of freedom, implemented using PyTorch [57]. In item 0, we first define the constraints for each state variable and time evolution on the computational graph. In item 1, define the geometric constraints for each system. This is the only part that needs to be rewritten when the system changes. In item 2, the partial Lagrangian is defined on the computational graph. The equations of motion with friction are used here to make the behavior easier to understand in the sample graph. In item 3, the constraints of the equations of motion are defined on the computational graph as edited differential equations using partial Lagrangians. Item 4 performs the backpropagation of the computed graph. In item 5, finally, the dynamics-aware inertia term $I$ is obtained as a coefficient of $\ddot{\theta}$ : acc, and the state variables updated by backpropagation are stored as internal variables.

As a precaution, since the Lagrangian is an edited differential equation, the Lagrangian must be registered in the computational graph as functions of $\theta$, $\omega$, and $t$. Therefore, the time evolution constraint is written in the program so that the variable theta is a function of omega and dt.

The computational graph generated by this sample is shown in Figure A1. In practice, programmers can analyze partial differential equations automatically without dealing directly with this complex graph, but only by writing algorithms similar to the sample code.

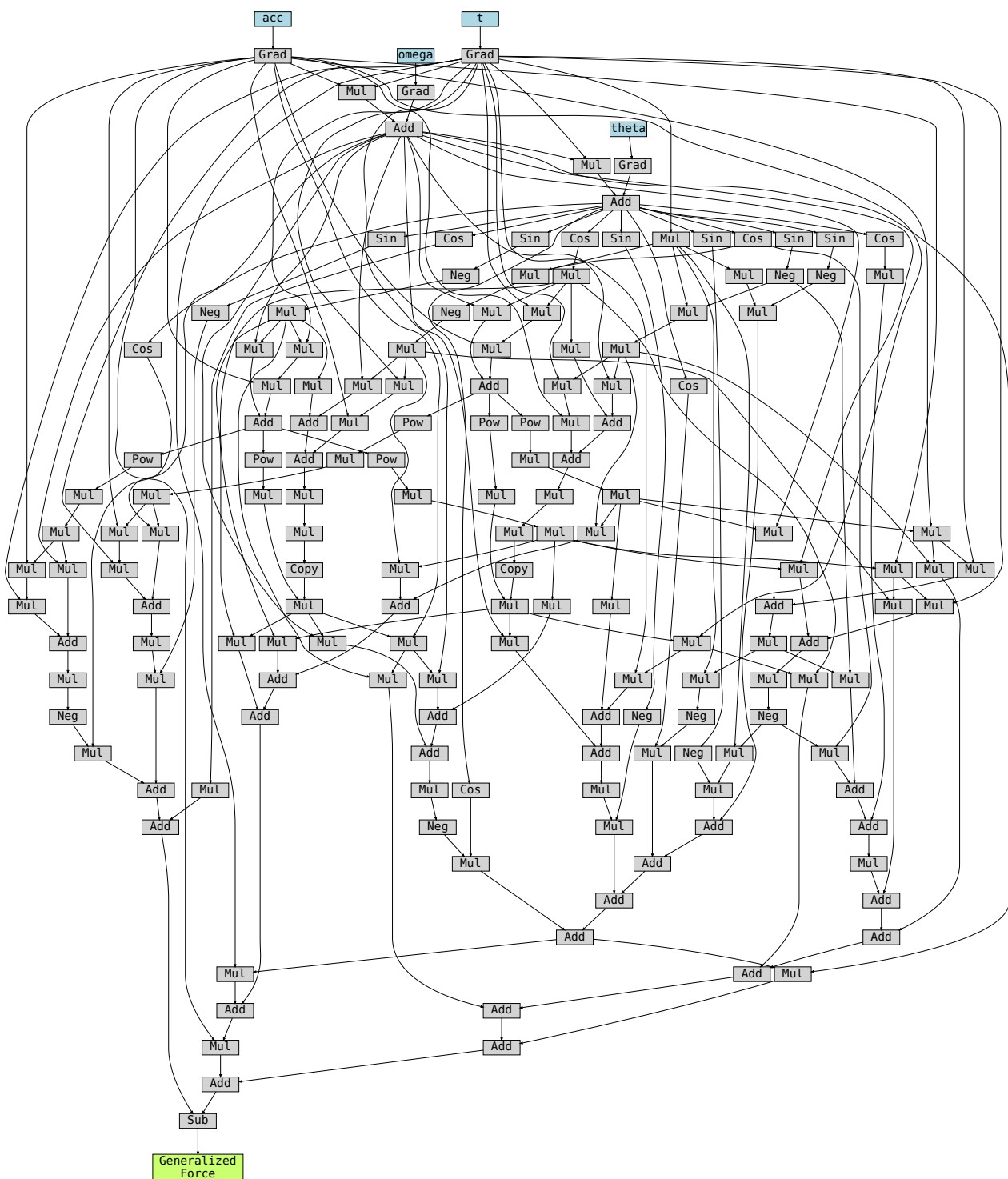

**Figure A1.** Example of a generated calculation graph of a partial Lagrangian (reading downward: forward calculations as inverse dynamics; reading upward: backward calculations as dynamics).

**Listing A1.** Example of the partial Lagrangian module.

```python
# 0. State variables and their constraints of time evolution for AD
(_theta, _omega, _acc) = state
acc = torch.tensor(_acc, requires_grad=True)
omega = torch.tensor(_omega, requires_grad=True) + acc * self.dt
theta = torch.tensor(_theta, requires_grad=True) + omega *self.dt
# 1. Geometric constraints and their derivatives
x = self.l * torch.cos(theta)
y = self.l * torch.sin(theta)
dx = torch.autograd.grad(x, self.dt, create_graph=True)
dy = torch.autograd.grad(y, self.dt, create_graph=True)
# 2. (Partial) Lagrangian with friction
K = 0.5*self.m * (dx[0]**2+dy[0]**2)
P = 0.5*self.k * (x**2+y**2) + self.m*9.8*y
R = 0.5*self.c * (dx[0]**2+dy[0]**2)
L = K−P
# 3. Lagrange Equation of Motion
dLdth = torch.autograd.grad(L, theta, create_graph=True)
dLdomega = torch.autograd.grad(L, omega, create_graph=True)
dLdt = torch.autograd.grad(dLdomega, self.dt, create_graph=True)
dRddth = torch.autograd.grad(R, omega, create_graph=True)
Iacc = dLdth[0] − dRddth[0]
T = dLdt[0] − Iacc # balance
# 4. Backpropagation
T.backward(retain_graph=True)
# 5. Convert force to acc
I = acc.grad
acc = float(Iacc/I)
omega += acc * self.dt
state = (theta, omega, acc)
return state
```

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
