# Peer review of "Partial Lagrangian for Efficient Extension and Reconstruction of Multi-DoF Systems and Efficient Analysis Using Automatic Differentiation"

_robotics, doi:10.3390/robotics11060149_

Round 1

Reviewer 1 Report

In this manuscript, a partial Lagrange method that can handle the Lagrange equation efficiently has been proposed by using a divide-and-conquer approach. It is claimed the proposed method can be used for manual derivation and benefits from automatic differentiations. The topic addressed in the paper can be of interest to the research community working in the area of multi-body dynamics. However, it is felt that the novelty of the work is not coming out clearly. The paper requires minor revision before it may be considered for a journal. The following comments are attempted to help the authors in improving the paper:

Literature survey requires thorough revision. It should critically discuss the existing work and clearly state how the present work is different from the existing work. Please clearly state novelty of the proposed work and provide salient contributions of the work in the form of bullets.

In Section 2, the procedure for automatic differentiations has been presented. However, the required operations are not mentioned. Why the characteristics concerning the computational efficiency of the algorithm are not discussed?

Several similar theories and derivational processes can be found in the following papers. What is the difference and connection between them? If there are some correlations between them, I think it’s necessary to reference them and highlight the innovation point of this manuscript.

ü  M. Poursina, K.S. Anderson, "An Extended Divide-And-Conquer Algorithm For a Generalized Class of Multibody Constraints," Multibody System Dynamics, DOI 10.1007/s11044-012-9324-9, 2012.

ü  J.H. Critchley, A. Binani, and K.S. Anderson, "Design and Implementation of an Efficient Multibody Divide and Conquer Algorithm", ASME Journal of Computational and Nonlinear Dynamics, Vol. 4(2) March 2009

ü  R.M. Mukherjee, K.D. Bhalerao and K.S. Anderson " A Divide and Conquer Direct Differentiation Approach for Multibody Systems Sensitivity Analysis ", Structural and Multidisciplinary Optimization, Available online at http://www.springerlink.com, Paper DOI10.1007/s00158-007-0142-2, 2007.

Authors should specify the information on the computers used and the computational time needed to solve the several cases.

As the respected authors approve, there are several approaches to avoid computing the Lagrange multipliers associated with the constraints. For example, recently a scholar has proposed recursive Gibbs-Appell formulation to derive the motion equations of different robotic systems with less computational complexity. As a consequence, it should be mentioned in introduction section to provide insights about the computational efficiency.

ü  A Zahedi, AM Shafei, M Shamsi, On the dynamics of multi-closed-chain robotic mechanisms, International Journal of Non-Linear Mechanics 147, 104241

Reviewer 2 Report

1.  The novelty of the proposed scheme should be highlighted carefully. Moreover, the importance of the suggested method should be further addressed.

2. The literature review is not sufficient. To help authors in this direction, I suggest the following references:

https://doi.org/10.1017/S0263574719000080

3. The conclusions should be extended and future lines of research should be discussed with more care. Also, conclusions should be supported by some data.

Reviewer 3 Report

See an attached sheet.

Reviewer 4 Report

This paper propose a partial Lagrange method that can handle the Lagrange equation efficiently even for multi-degree-of-freedom systems by using a divide-and-conquer approach. The paper is fairly well documented, and the mathematical developments seem to be correct. Here are some comments 

1. This paper uses the equation of motion with the Lagrangian. What is superiority?

2. How to understand the practical meaning of inequalities (5) and (6)?

3. Why the authors choose the link end position and its velocity? Any other choices?

4. Do you have comparisons with other Lagrange methods for multi-degree-of-freedom systems? Please discuss.

5. How do you actually ensure the stability? What happens if quite different initial conditions are accounted for?

6. The practical insights of the proposed methods are suggested to be discussed in order to highlight the significance and the novelty.

Round 2

Reviewer 2 Report

the current version can be published. 

Reviewer 3 Report

There are no specific comments.